# A Comprehensive Systematic Review Coupled with an Interacting Network Analysis Identified Candidate Genes and Biological Pathways Related to Bovine Temperament

**DOI:** 10.3390/genes15080981

**Published:** 2024-07-25

**Authors:** Gilberto Ruiz-De-La-Cruz, Thomas H. Welsh, Ronald D. Randel, Ana María Sifuentes-Rincón

**Affiliations:** 1Laboratorio de Biotecnología Animal, Centro de Biotecnología Genómica, Instituto Politécnico Nacional, Reynosa 88710, Mexico; gruizd1800@alumno.ipn.mx; 2Department of Animal Science, Texas A&M University, College Station, TX 77843, USA; thomas.welsh@ag.tamu.edu; 3Texas A&M AgriLife Research, Overton, TX 75684, USA; ron.randel@ag.tamu.edu

**Keywords:** temperament pathways, *SST*, Kelch family genes, cattle, behavior, splicing, ubiquitination

## Abstract

Comprehension of the genetic basis of temperament has been improved by recent advances in the identification of genes and genetic variants. However, due to the complexity of the temperament traits, the elucidation of the genetic architecture of temperament is incomplete. A systematic review was performed following the Preferred Reporting Items for Systematic Reviews and Meta-Analyses (PRISMA) statement to analyze candidate genes related to bovine temperament, using bovine as the population, SNPs and genes as the exposure, and temperament test as the outcome, as principal search terms for population, exposure, and outcome (PEO) categories to define the scope of the search. The search results allowed the selection of 36 articles after removing duplicates and filtering by relevance. One hundred-two candidate genes associated with temperament traits were identified. The genes were further analyzed to construct an interaction network using the STRING database, resulting in 113 nodes and 346 interactions and the identification of 31 new candidate genes for temperament. Notably, the main genes identified were *SST* and members of the Kelch family. The candidate genes displayed interactions with pathways associated with different functions such as AMPA receptors, hormones, neuronal maintenance, protein signaling, neuronal regulation, serotonin synthesis, splicing, and ubiquitination activities. These new findings demonstrate the complexity of interconnected biological processes that regulate behavior and stress response in mammals. This insight now enables our targeted analysis of these newly identified temperament candidate genes in bovines.

## 1. Introduction

Bovine temperament is defined as the animal’s reaction to human presence or manipulation. Temperament is a moderately heritable trait upon which selection pressure can be applied to improve animal welfare and economically relevant traits [1,2]. Animals with more excitable temperament display lower body weight and weaning weights, and the selection of docile animals could help ensure animals with optimal weights [3]. Quality characteristics are related to bovine temperament; animals had a negative correlation between temperament and rib thickness and carcass weight [4]. Due to temperament’s importance in cattle production, several well-standardized tests to score temperament have been developed, and all are aimed at quantifying bovine reactions to human interaction [5]. In unrestricted tests, such as the open field test, the cow is free to move within a test area and evaluate a novel environment. In such cases, the human observer can identify reactivity to novelty and social isolation [6]. Restricted tests are associated with animal stress by reduced free motion, such as the chute score, which evaluates the number of movements as part of the temperament selection trait [7]. The exit velocity is another restricted evaluation, quantifying the rate of speed at which the animal travels at 1.8 m when it is released [8]. Due to the correlations between cortisol concentrations and exit velocity, this measure of temperament was proposed as an indication of physiological stress responses of animals’ future encounters with humans [9].

At the genomic level, bovine temperament studies have evolved from single-gene strategies to genome-wide association studies (GWASs) [10,11,12]. To explore the interaction between genetic, phenotypic, and environmental factors, it is crucial to obtain a more complete view of the genetic basis of bovine temperament.

Identifying gene interactions is necessary to help understand biological pathways and functions linked to bovine temperament. Analysis of genetic interactions could reveal key pathways and putative genes that modulate bovine temperament, providing insights for designing bovine breeding and management strategies [13]. Gene interaction has been a tool to support temperament analysis, as reported by Paredes-Sánchez et al. [12]. For example, Lindholm-Perry et al. [14] reported that *GRIA2* mediated interactions of the genes *CACNG4* and *NRXN3*. Knowing the interactions between genes will help understand and define the pathways in which temperament develops since there are few studies that integrate gene interaction. Progress in the identification of interaction pathways related to temperament has been favored by the common characteristics shared by genes in bovine and human temperament that cause behavioral disorders by similarities in orthology and ontology [15]. Most of the reported studies highlight pathways related to the hypothalamic–pituitary–adrenal (HPA) axis system as the principal response to stressful stimuli [16,17]; however, the complexity of the temperament may involve additional biological pathways.

This review applies the current knowledge about gene variations and their effect on bovine temperament to elucidate the interaction network of new candidate genes and pathways that describe the genetic architecture of this important trait.

## 2. Materials and Methods

### 2.1. Systematic Review

Developing a systematic review requires a strategy that reduces search bias. The Preferred Reporting Items for Systematic Reviews and Meta-Analyses (PRISMA) statement provides guidance and support for comprehensive and transparent research reporting [18].

Using the PRISMA statement [19], a search was conducted in Google Scholar (https://scholar.google.com/; access date: 6 February 2024), PubMed (https://pubmed.ncbi.nlm.nih.gov/; access date: 6 February 2024), and Web of Science (https://www.webofscience.com/wos/woscc/basic-search; access date: 6 February 2024) databases. The categories of population, exposure, and outcome (PEO) components were used to define the scope of the literature search [20]. The search was designed to find at least one source for each category and was accomplished using references from other systematic reviews for bovine temperament [2,21]. The population category was defined by “Bos*”, “cattle”, or “bovine” terms; the exposure category was defined by “SNP”, “Gene*”, and “candidate gene*” terms; and finally, “temperament” was used for the outcome category, with “Chute Score” or “Crush Score” or “Exit Velocity” or “Pen Score” or “Temperament Score” or Flight* or Aggressi* or Excit* or Docile or Reactive or Temperamental terms.

The first step of the workflow was quality control to remove duplicate records from the database. Unique records underwent three selection steps: (1) title analysis that aimed to discard records based on human and non-*Bos taurus* models and remove incomplete files, doctoral or master’s degree theses, congress memories, and poster summaries; (2) abstract information analysis to discard those records that explicitly declared no genetic association of the studied SNP; and (3) resulting records were thoroughly analyzed to find and list the genes previously associated with bovine temperament. The reviews or systematic review articles were not excluded because their integration increases the search quality, and their references could give information about studies not included in our search.

The list of candidate genes selected for the subsequent analysis included those genes harboring single SNPs, SNP haplotype blocks, or quantitative trait loci with a reported effect on temperament traits, and it also included candidate genes with no reported effect on a temperament trait.

### 2.2. Development of Interaction Network and Gene Enrichment 

The literature search result generated a list of temperament candidate genes with 102 genes that encode proteins (Appendix A). STRING [22] (https://string-db.org/; access date: 26 February 2024) is a database of known and predicted protein–protein interactions used to develop an interaction network through the search for “Multiple proteins”. This is a tool that evaluates the interaction of proteins according to physical and functional associations from sources such as genomic context, high-throughput experiments, co-expression, and the literature. The network was clustered through the Markov clustering algorithm option with the default inflation parameter [23]. The interaction network generated information enriched by the Kyoto Encyclopedia of Genes and Genomes and was used to identify principal pathways for temperament candidate genes.

## 3. Results

The search query yielded 565 results, comprising 509 from Google Scholar, 12 from PubMed, and 44 from Science of Web. Adhering to the PRISMA workflow, 49 duplicate results were removed (Appendix A). The screening process initially focused on article titles to exclude non-*B. taurus* or non-*Bos indicus* results and non-article entries. A total of 458 results were eliminated after reviewing the article titles. The screening process continued with abstract screening, resulting in the removal of 22 articles. In total, 36 articles were assessed and included in this review based on the candidate genes criteria (Figure 1).

From the 36 articles of the systematic review search, 102 candidate genes for temperament traits were found. The principal temperament tests were cortisol concentration, exit velocity, flight speed, flight time, novel object, and milking temperament. Some emotion circuits of temperament tests were fear, panic, and seeking; others reported independently were aggressiveness, general temperament, and nervousness.

### 3.1. Establishing the Gene Interaction Network

From a list of 102 selected genes, STRING found interactions for 31 new candidate genes (Table 1). The interaction network showed 113 nodes with 346 edges; the average node degree was 6.12, the clustering coefficient was 0.615, and the protein–protein interaction enrichment *p*-value was <1.0 × 10^−16^. The main annotated keywords were phosphoprotein (FDR = 5.41 × 10^−6^), cytoplasm (FDR = 6.55 × 10^−6^), catecholamine biosynthesis (FDR = 1.08 × 10^−5^), Ubl conjugation pathway (FDR = 1.08 × 10^−5^), disulfide bond (FDR = 5.28 × 10^−5^), kelch repeat (FDR = 5.38 × 10^−5^), mRNA splicing (FDR = 5.38 × 10^−5^), cytoplasmic vesicle (FDR = 5.38 × 10^−5^), and nucleus (FDR = 5.38 × 10^−5^).

### 3.2. Gene Clusters and Pathways

The gene interaction network showed clusters sharing biological functions such as AMPA, hormones, neuronal maintenance, protein signaling, neuronal regulation, serotonin synthesis, splicing, and ubiquitination activities. These clustered genes were described according to the metabolic pathways identified and described through the Kyoto Encyclopedia of Genes and Genomes database.

The AMPA (α-amino-3-hydroxy-5-methyl-4-isoxazole propionic acid) receptors (AMPA cluster) integrate the new candidate gene of calcium voltage-gated channel auxiliary sub-unit γ 7 (*CACNG7*). The AMPA cluster displays an interaction with the glutamate pathway that is involved in functions as a protein-modifying transmembrane AMPA-glutamate receptor (Figure 2).

The hormones cluster included five new candidate genes (*CLU*, *PCSK1*, *PPY*, *PYY*, and *SST*) (Figure 2), and their principal pathway participation was the neuroactive ligand–receptor interaction. The genes for pancreatic polypeptide (*PPY*) and peptide YY (*PYY*) have gastrointestinal functions and act at the hypothalamus to modulate appetite, food intake, and energy balance (Figure 3A). Somatostatin (*SST*) inhibits the hormonal release to indirectly reduce appetite (Figure 3A). The *CLU* gene works in apoptosis, cellular stress response, and lipid transport and metabolism. The *PCSK1* gene is involved in hormone processing (e.g., *POMC* and somatostatin). Specifically, the cluster displaying hormone processing is through the corticotropin-releasing hormone pathway (Figure 3B). 

The neuronal maintenance cluster revealed the nuclear receptor subfamily 4 group A member 2 (*NR4A2*) as a new candidate gene. This is a nuclear receptor that functions as a transcription factor associated with neuronal development and maintenance and survival of dopaminergic neurons involved in movement. It also functions as a neuroprotector in response to injuries in the nervous system. A loss of function of the *NR4A2* gene is associated with Parkinson’s disease (Figure 4A,B). Another cluster involved with neuronal activities was the neuronal regulation cluster that is linked with the engulfment and cell motility 2 and 3 genes (*ELMO2* and *ELMO3*). *ELMO2* and *ELMO3* interact as Rac activators. Rac is a member of the Rho family of small GTPases that regulate the differentiation of axons and dendrites (Figure 4C). According to the KEGG database information, both clusters show coincidence in the neuronal activity pathways.

The protein signaling cluster was integrated by the calreticulin gene (*CALR*), a calcium-binding chaperone that promotes folding, assembly, and quality control in the endoplasmic reticulum (ER). The signal transducer and activator of transcription 3 gene (*STAT3*) works as a signal transducer and transcription activator mediating cellular responses to growth factors. The *STAT3* activation recruits coactivators to the promoter region of a target gene. The E3 ubiquitin-protein ligase gene (*TRIM11*) promotes the degradation of insoluble ubiquitinated proteins (Figure 5). The String database does not define a specific pathway for these genes; the protein signaling pathway was determined based on information on gene activities reported in the KEGG database.

The neurotransmitters cluster did not include new candidate genes; however, multiple interactions were found with other clusters. The principal activity of the neurotransmitters cluster was the neuroactive ligand–receptor interaction (Figure 6A,B). The cluster includes genes that function through enzymes (syntheses, sub-production, or degradation), transport, reception, and precursors of neurotransmitters. Within the neurotransmitters cluster, we found the serotonin synthesis cluster (*AKR1B1*, *TDO2*, *TPH1*, and *TPH2* genes) being the folate biosynthesis the main pathway displayed. The serotonin synthesis cluster included the new candidate gene *AKR1B1*, an aldose reductase, which catalyzes NADPH activity towards aromatic and aliphatic aldehydes, ketones, monosaccharides, bile acids, and xenobiotics substrates. Also, *AKR1B1* catalyzes the reduction of glucose to sorbitol during hyperglycemia (Figure 6C). 

The splicing cluster (Figure 7) involved genes with activity in the RNA transport pathway, specifically *CASC3* (cancer susceptibility candidate 3), a component of the exon junction complex (EJC) involved in subsequent steps of splicing, such as export from the nucleus to the cytoplasm and translation. The Gemin complex genes (*GEMIN6*, *GEMIN7*, and *GEMIN8*) and survival of motor neuron 2 (*SMN2*) function in the biogenesis and assembly of spliceosome components such as small nuclear ribonucleoproteins (snRNPs). The small EDRK-rich factor 1A gene (*SERF1A*) is involved in protein homeostasis and response to proteotoxic stress. The serine/threonine kinase receptor−associated protein (*STRAP*) influences TGF-β signaling. PGC−1− and ERR−induced regulator in muscle 1 (*PERM1*) may affect energy metabolism due to its effects on fatty acid oxidation and cellular respiration. Additionally, this pathway includes aspects of exploration, foraging, and social interaction.

The ubiquitination cluster had the most new candidate genes (Figure 8). The interaction network, according to the KEGG database, revealed that the pathway with the greater activity was protein degradation. The family of potassium channel tetramerization domain (KCTD) represented by the *KCTD13* and *KCTD5* genes work as associated subunits with the cullin 3 (CUL3) complex that allows the substrate selection for ubiquitination. Another family was that of the Kelch protein. It acts as an adapter subunit in the CUL3 complex that is involved in protein ubiquitination and vesicle trafficking regulation and signaling (*KLHL12*, *KLHL20*, and *KLHL9*) and protein degradation (*KLHL13*, *KLHL21*, and *KLHL3*). The *SPOP* and *UBE2M* genes are associated with the adapter subunit for E3 and E2 ubiquitin ligases of the CUL3 complex, which functions in the regulation of proteins in conjunction with protein ubiquitination.

## 4. Discussion

As previously described, the PRISMA framework is a set of guidelines or steps to systematically search papers and literature for review-based studies [18]. The use of a systematic review to justify and inform the design of a new study and to place new results in context is a process called evidence-based research (EBR). Here, we incorporate both approaches: using the systematic review as a first step to generate data for mining and then employing an interaction network approach to identify new candidate genes related to bovine temperament. Based on their interactions, we describe the biological pathways associated with this trait.

The systematic review approach allowed the identification of 31 new candidate genes for bovine temperament. From these, the candidate genes showing the greater interactions were *SST* (18 interactions) and Kelch−like family members (*KLHL3*, *9*, *12*, *13*, *20*, and *21*; 10 interactions each). Also, eight clusters (AMPA, hormones, neuronal maintenance, protein signaling, neuronal regulation, serotonin synthesis, splicing, and ubiquitination) were identified, which allowed a description of their interactions and involvement in biological pathways.

The *SST* gene is an important participant in biological pathways such as stress response, specifically the control of post-stress anxiety relief by activating the reward system with self-grooming [24]. In mammals, the population of interneurons γ-aminobutyric acid (GABA) that express *SST* controls the feedback of glutamatergic projection neurons [25]. The decreased function of cells expressing *SST* together with GABA induces depression or anxiety-like behaviors [26]. Also, *SST* is included in the hormone pathway that inhibits the expression of growth hormone (GH) mRNA and GH secretion [27].

The Kelch−like family members have been reported to function as subunit adapters in the CUL3 complex and are involved in synaptic responses [28]. The Kelch family genes function in protein ubiquitination with non-degradative activities that regulate vesicle size and promote collagen transport (*KLHL12*), promote trafficking of post-Golgi bodies (*KLHL20*), promote cytokinesis (*KLHL9*, *KLHL13*, and *KLHL21*), and as protein degradation (*KLHL3*) [29]. Other activities of the Kelch family genes include the modulation of voltage-dependent calcium channels and increasing the excitability of POMC neurons [30].

The following sections discuss eight identified clusters, their interactions, and related biological pathways.

### 4.1. AMPA Cluster

Glutamate is the major excitatory neurotransmitter in the nervous system and is associated with reactions to stressors. The ionotropic glutamate AMPA receptor subunit is encoded by the *GRIA2* gene [31]. The AMPA receptor formation requires the γ-calcium subunits *CACNG4* and *CACNG7* [32]. The *CACNG4* and *GRIA2* genes have been associated with bovine temperament in different studies and populations [14,33,34].

### 4.2. Hormones Cluster

In our analysis, the KEGG database referenced the neuroactive ligand–receptor interaction as the principal pathway for the functions of the hormonal cluster gene. This pathway is widely studied due to its role in hormone release influenced by stress and neuroactive response, specifically by cortisol production, which is a parameter for measuring bovine temperament [17].

The HPA axis response to stress is activated through neurotransmitters, such as GABA and glutamate [17]. The plasma concentration of ACTH is related to bovine temperament, as ACTH controls cortisol concentrations [9,35]. ACTH, derived from the *POMC* gene, is the principal hormone linked with this cluster. *POMC* gene function is facilitated by the *PCSK1* gene along with *CPE*. These genes are activated in response to stressors and affect appetite and energy metabolism. The *POMC* gene was associated with the flight−or−fight response with allelic change in two SNPs in the 3′ untranslated region (UTR) and has been associated with temperament using the Pen Score test [36]. Additionally, *CPE* expression is activated during physiological and environmental stressful conditions [37]. Feeding behavior, influenced by hormones for appetite reduction, involves the hormone processing of somatostatin (*SST*) by *PCSK1*, related to α-MSH by ACTH, and anxiolytic stress mediator *NPY* that modulates feeding behavior [38]. It has been found that somatostatin (*SST*) inhibits growth hormone (*GH1*) regulated by *POU1F1*, which is involved in appetite regulation by the *LEP* gene [39,40].

### 4.3. Neuronal Maintenance Cluster

The neuronal synapse pathway revealed genes with functions for transcription factors (*BARHL2*), neuronal development and maintenance (*NR4A2*), synaptic function (*TOPAZ1*), and vesicle transport and the regulation of neuronal growth (*KIFAP3*) [41].

### 4.4. Protein Signalization and Ubiquitination Clusters

In the protein signaling and ubiquitination pathways, the genes *TRIM11* [42] and *UBE2M* [43] participate through the ubiquitin–proteasome system for protein degradation and E3 ligases for protein ubiquitination. Ubiquitination and protein degradation are stress-inducible functions [44]. The *TRIM11* gene has a role in protection against tauopathies, and downregulation could contribute to behavioral problems [45]. Additionally, the TRIM family genes have been related to the regulation of the nuclear factor of activated T cells 4 (*NFATC4*) for neuronal apoptosis [46].

The CUL3 complex, together with the genes of the KCTD family and the *SPOP* gene, participates in the regulation of proteins through ubiquitination [47]. Members of the KCTD family have been linked to GABA-signaling pathways and multiple diseases [48]. Autism and schizophrenia have been associated with *KCTD13* deletions, and obesity was associated with *KCTD15* variants [49]. The KCTD family is also involved in the control of cyclic adenosine monophosphate (cAMP) signaling required for protein kinase A-mediated protein phosphorylation’s involvement, increasing recognition sites for the E3 ligase enzymes responsible for ubiquitination [50]. In addition, the Kelch protein family genes act as adaptor subunits, regulating vesicle trafficking, protein degradation, and modulating voltage-gated calcium channels, thereby increasing the excitability of neurons [30].

Activities in organelles such as the mitochondria require integration with the core component of the E3 ubiquitin ligase complex (*SIRT3*), allowing for the deacetylation of protein [51]. Endoplasmic reticulum (ER) stress could result in brain injury [52]. Neuronal survival under ER stress requires the integration of the *CALR* gene that encodes an ER chaperone molecule [53]. The protein regulation pathway is important for the maintenance of redox reactions by reversible oxidation (*TXN* and *TXNDC8*), which supports metabolic homeostasis [54,55].

### 4.5. Regulation Neuron Cluster

Genes such as *ELMO2*, *ELMO3*, and *DOCK1* may mediate the relationship between stress factors and temperament by activation of Rac. This leads to the reorganization of the actin cytoskeleton, possible cell migration, and receptor mediation [56].

### 4.6. Serotonin Synthesis Cluster

Serotonin synthesis genes influence the functioning of neurotransmitters in the neuroactive ligand–receptor interaction pathway. The *AKR1B1* gene displayed participation in serotonin biosynthesis alongside the *TPH1* and *TPH2* genes that participate in the production of enzymes important in serotonin production [57]. Brain tissue damage due to inflammation and excessive cellular apoptosis has been related to the increase in the expression of the *AKR1B1* gene. Inhibition of the *AKR1B1* gene is a strategy for neuroprotection [58].

Another involvement of the *AKR1B1* gene is together with *ALDH5A1*, through the conversion of aldehydes to alcohols, eliminating toxic aldehydes produced from GABA metabolism, protecting cells from oxidative damage by reducing the levels of reactive oxygen molecules [59]. The *AKR1B1* gene has been reported as a hypomethylated gene located within a CpG island and as hypermethylated DNA located within an intron region in prenatally stressed Brahman calves [60].

### 4.7. Splicing Cluster

The splicing pathway was highlighted by the varied involvement of genes in different activities of the process. For example, the *SNRPF* gene is involved in RNA processing and splicing functions and has been associated with circadian rhythm and autism spectrum disorder in humans [61]. Upregulation of *SNRPF* gene expression is related to axotomy regeneration and degradation of UBE family proteins with ubiquitin functions [62]. The *CASC3* gene encodes a protein that requires an activation signal triggered by the exon junction complex (EJC) of the messenger ribonucleoprotein (mRNP) that binds to mRNAs. The CASC3 protein shuttles between the nucleus and the cytoplasm but is not an obligate component of all EJCs [63]. Gemin genes (6–8) are part of the cytoplasmic survival of motor neuron complex [64], which is essential for structural rearrangements of small nuclear RNA (snRNAs) during maturation by facilitating ATP-driven structural changes in snRNAs that expose the Sm site allowing binding to Sm protein [65].

The *PERM1* gene showed activity in the regulation of mitochondrial biogenesis [66,67] and the regulation of mitochondrial oxidative capacity [68]. The *STRAP* gene has been identified as a possible spliceosome-associated factor since the loss of function brings with it alternative splicing events. In addition, *STRAP* participates in the assembly of snRNP proteins [69]. Also, *PERM1* and *STRAP* have been reported in mammary gland infection regulated by *bta-miR-145* [70]. The survival motor neuron (SMN) protein is dependent on the expression of the *SMN2* and *SERF1A* genes. Both genes could work in alternative splicing [71,72]. 

## 5. Conclusions

Applying a novel approach to identifying candidate genes, we revealed and confirmed the complexity and interconnections of biological processes that regulate temperament and stress response in mammals.

Thirty-one novel genes were revealed as potential candidate genes influencing the expression of temperament in cattle. Genes such as *SST* and the Kelch family showed important biological interactions and functional connections in different aspects of temperament expression.

This facilitates observation of the connections to the most studied pathways, such as neurotransmission of the HPA axis with feeding behavior and metabolic activities such as protein ubiquitination. Temperament is a very complex trait affected by multiple genes and gene interactions. This information is important for future efforts to understand temperament’s influence on animal well-being and productivity.

## Figures and Tables

**Figure 1 genes-15-00981-f001:**
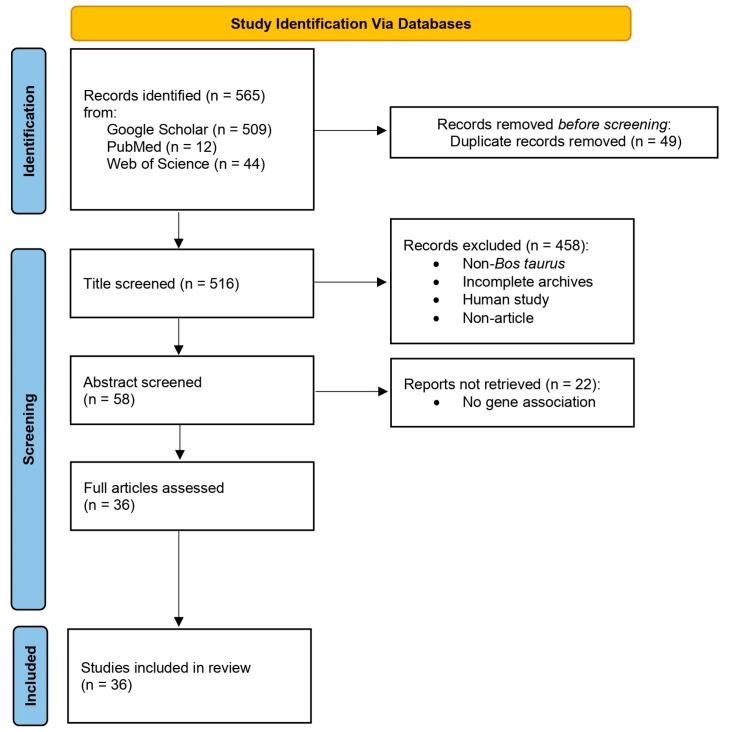
Workflow for selection of results returned using the PRISMA protocol 2020.

**Figure 2 genes-15-00981-f002:**
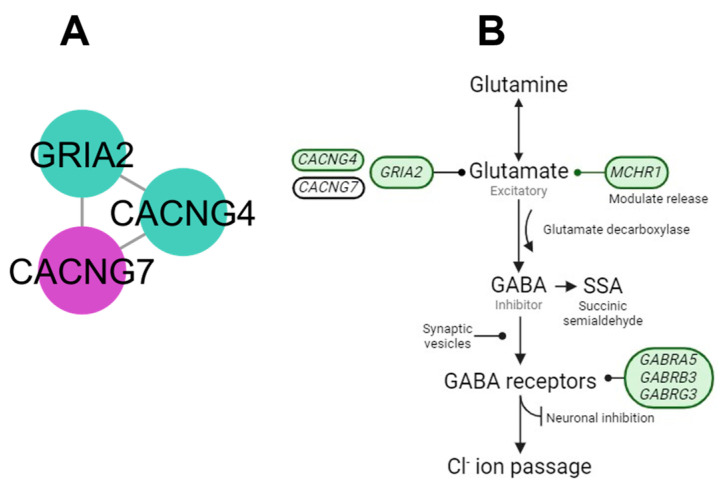
Participatory genes of the glutamine process pathway. (**A**) Interaction network among subunit γ calcium with AMPA subunit genes and new candidate gene (purple). (**B**) Glutamine-to-glutamate process describing the interaction of the AMPA cluster for reception of glutamate molecules with the new candidate gene (white).

**Figure 3 genes-15-00981-f003:**
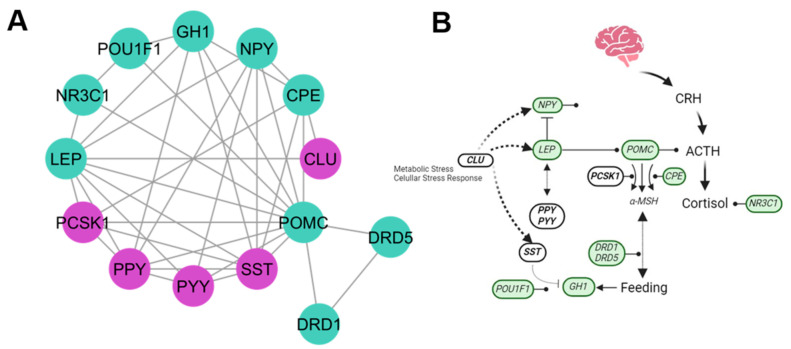
Hormonal process for feeding behavior. (**A**) The hormones cluster shows the new candidate genes (purple) and genes previously related to bovine temperament (green). (**B**) Interaction pathway of new candidate genes (white), interacting as inhibitors of growth, feeding modulation, and enzymes for processing of hormones.

**Figure 4 genes-15-00981-f004:**
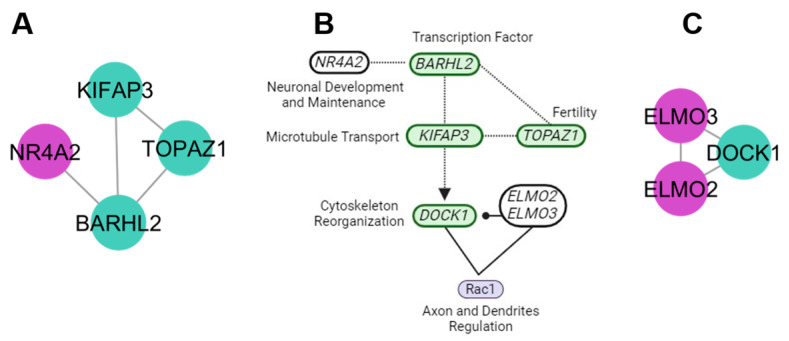
Clusters involved in neuronal processes. (**A**) Neuronal maintenance cluster, with new candidate gene (purple) and genes related to bovine temperament (green). (**B**) Principal activities of genes for neuronal maintenance, development, regulation, and differentiation. (**C**) Neuronal regulation cluster, with new candidate genes (purple) and genes related to bovine temperament (green).

**Figure 5 genes-15-00981-f005:**
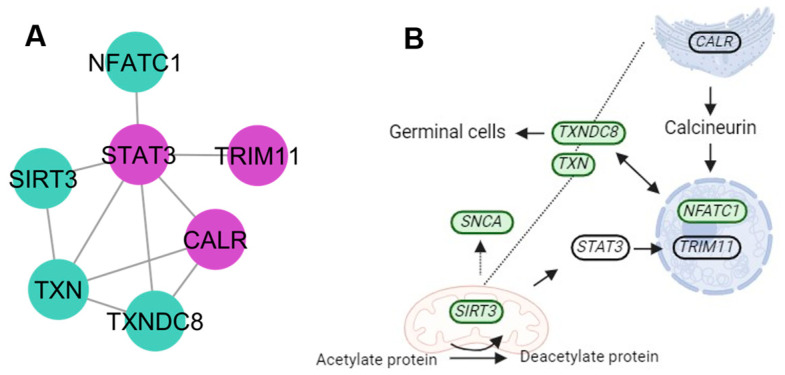
Protein signaling pathways of new candidate genes. (**A**) Interaction network between new candidate genes (purple), with genes related to bovine temperament (green). (**B**) The participation of genes functioning through the protein signaling pathways involves different organelles, such as the nucleus, mitochondria, and rough endoplasmic reticulum.

**Figure 6 genes-15-00981-f006:**
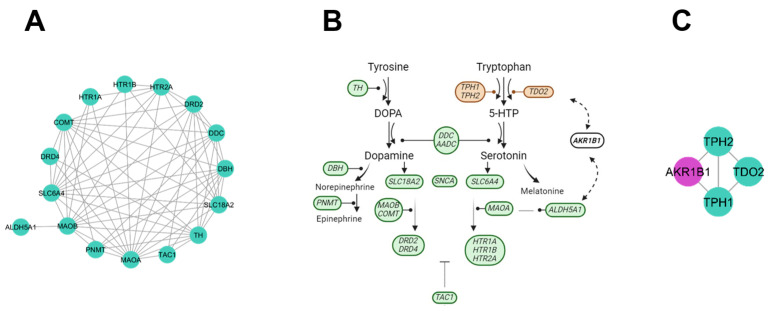
Dopamine and serotonin production genes. (**A**) Neurotransmitters cluster. (**B**) Dopamine production genes (green), serotonin genes (brown), and not previously reported interacting genes (white). (**C**) Serotonin synthesis cluster with the new candidate gene *AKR1B1* (purple).

**Figure 7 genes-15-00981-f007:**
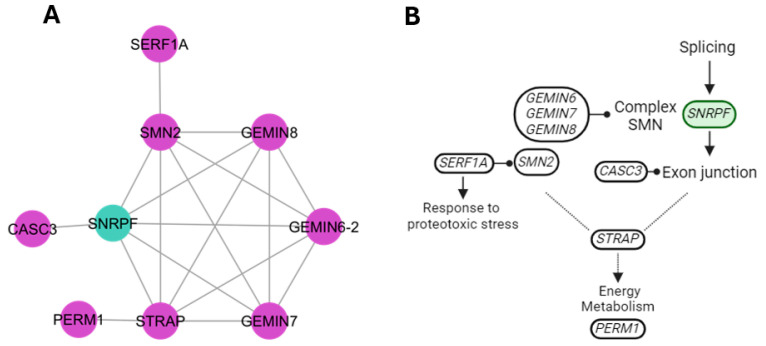
Participation of new candidate genes in splicing pathways. (**A**) New candidate genes (purple) interact around the *SNRPF* gene. (**B**) Representation of the participation of splicing genes and their relationship to stress and metabolism.

**Figure 8 genes-15-00981-f008:**
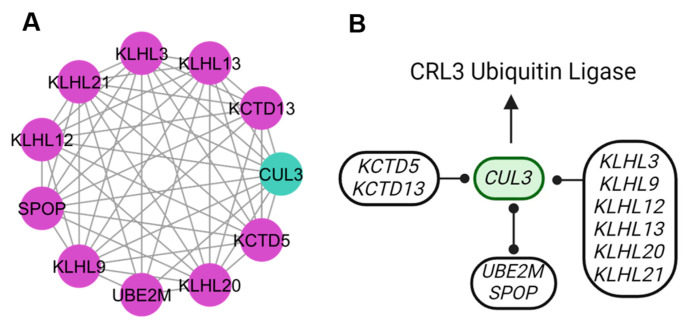
Genes for the ubiquitination process. (**A**) The ubiquitination cluster shows the new candidate genes (purple). (**B**) Interactions of the new candidate genes with *CUL3* gene are part of a different function for the CUL3 complex.

**Table 1 genes-15-00981-t001:** New candidate genes for bovine temperament.

Group	Gene	Name	Biological Function
Enzyme	*AKR1B1* ID:317748	aldo-keto reductase family 1, member B1 (aldose reductase)	Carbohydrate metabolism. Catalyzes the reduction of glucose to sorbitol during hyperglycemia. Pentose and glucuronate interconversions, fructose and mannose metabolism. Reduces steroids and their derivatives and prostaglandins and participates in folate biosynthesis.
	*PCSK1* ID:281967	proprotein convertase subtilisin/kexin type 1	Involved in hormone and protein processing. Substrates include proopiomelanocortin (POMC), renin, enkephalin, dynorphin, somatostatin, insulin, and agouti-related protein (AGRP).
	*UBE2M* ID:613343	ubiquitin-conjugating enzyme E2 M	Ubiquitin-mediated protein degradation, cell cycle regulation.
Protein	*CACNG7* ID:539969	calcium voltage-gated channel auxiliary subunit γ 7	Regulation of voltage-gated calcium channel activity.
	*CALR* ID:281036	calreticulin	Protein processing and folding in the endoplasmic reticulum, endoplasmic reticulum stress response, regulation of calcium homeostasis.
	*CASC3* ID:531673	CASC3 exon junction complex subunit	Regulation of messenger RNA splicing, exon junction complex (EJC).
	*CLU* ID:280750	clusterin	Apoptosis, cellular stress response, lipid transport and metabolism.
	*ELMO2* ID:508361	engulfment and cell motility 2	Phagocytosis of apoptotic cells, Rho GTPase signaling.
	*ELMO3* ID:525427	engulfment and cell motility 3	Phagocytosis of apoptotic cells, cell migration, and cytoskeletal remodeling.
	*GEMIN6* ID:525263	gem nuclear-organelle-associated protein 6	Assembly of the small nuclear RNA (snRNP) complex, RNA biogenesis.
	*GEMIN7* ID:618024	gem nuclear-organelle-associated protein 7	Assembly of the small nuclear RNA (snRNP) complex, RNA biogenesis.
	*GEMIN8* ID:515968	gem nuclear-organelle-associated protein 8	Assembly of the small nuclear RNA (snRNP) complex, RNA biogenesis.
	*KLHL12* ID:768068	kelch-like family member 12	Ubiquitin-mediated protein degradation, vesicle formation.
	*KLHL13* ID:528138	kelch-like family member 13	Ubiquitin-mediated protein degradation, vesicular trafficking.
	*KLHL20* ID:511387	kelch-like family member 20	Ubiquitin-mediated protein degradation, response to hypoxia.
	*KLHL21* ID:506632	kelch-like family member 21	Ubiquitin-mediated protein degradation, cell cycle.
	*KLHL3* ID:533364	kelch-like family member 3	Potassium homeostasis, ion transport.
	*KLHL9* ID:767834	kelch-like family member 9	Ubiquitin-mediated protein degradation, cell cycle regulation.
	*PPY* ID:280900	pancreatic polypeptide	Regulation of appetite, energy metabolism.
	*PYY* ID:615800	peptide YY	Regulation of appetite, gastrointestinal motility.
	*KCTD13* ID:507911	potassium channel tetramerization domain containing 13	Regulation of ubiquitin-mediated protein degradation, GABA signaling.
	*KCTD5* ID:100125308	potassium channel tetramerization domain containing 5	Regulation of ubiquitin-mediated protein degradation, GABA signaling.
	*PERM1* ID:520080	PPARGC1- and ESRR-induced regulator, muscle 1	Regulation of mitochondrial metabolism, mitochondrial biogenesis.
	*STRAP* ID:510201	serine/threonine kinase receptor-associated protein	TGF-β signaling, regulation of cell growth.
	*SERF1A* ID:526395	small EDRK-rich factor 1A (telomeric)	Specifically unknown, but it may be involved in cellular stress processes.
	*STT* ID:280932	Somatostatin	Inhibition of hormone release, regulation of cell growth.
	*SPOP* ID:530618	speckle-type BTB/POZ protein	Ubiquitin-mediated protein degradation, signaling cell growth pathways.
	*SMN2* ID:281492	survival of motor neuron 2	RNA splicing, snRNP biogenesis.
	*TRIM11* ID:514580	tripartite motif-containing 11	Ubiquitin-mediated protein degradation, antiviral response.
	*NR4A2* ID:540245	nuclear receptor subfamily 4 group A member 2	Transcription of genes involved in neurogenesis, inflammatory response.
	*STAT3* ID:508541	signal transducer and activator of transcription 3	Cytokine signaling, cell proliferation and survival.

ID: Gene identification number of NCBI database for reference sequences.

## Data Availability

Data are available from the authors upon request.

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
