# Peer review of "A Comprehensive Systematic Review Coupled with an Interacting Network Analysis Identified Candidate Genes and Biological Pathways Related to Bovine Temperament"

_genes, 2024, doi:10.3390/genes15080981_

Round 1

Reviewer 1 Report

Comments and Suggestions for Authors

Dear Author,

 The objective of this study titled "A Comprehensive Systematic Review Coupled with an Interacting Network Analysis Identified Candidate Genes and Biological Pathways Related to Bovine Temperament" was to provide a literature review and interaction network analysis to identify candidate genes and some biological pathways associated with bovine temperament. The following suggestions aim to enhance the manuscript's readability.

 Minor comments:

      1.         Systematic Review and PRISMA Statement: The research adheres to the PRISMA (Preferred Reporting Items for Systematic Reviews and Meta-Analyses) guidelines, which is a systematic review framework designed to ensure rigor and transparency in literature searches. While the PRISMA system's methodology is not explicitly detailed in the article, its incorporation is laudable. To enrich the paper, an explanation of PRISMA's advancements and benefits should be seamlessly integrated into either the background introduction or the discussion section, elucidating how it enhances the study's methodology.

2.         Comprehensive Literature Search: The authors have meticulously searched multiple databases, including Google Scholar, PubMed, and Science of Web, bolstering the study's findings. To augment transparency, the specific search string utilized for the final query should be delineated in the methods section. Additionally, appending the comprehensive list of identified literature (encompassing 565 results) as a supplementary file would greatly facilitate reader verification and trust in the study's exhaustiveness.

3.         Identification of Novel Candidate Genes: The discovery of 102 candidate genes, with a subsequent network analysis revealing 31 putatively novel genes associated with temperament, marks a significant advancement. It is imperative to clarify the criteria defining 'novelty' in these genes, especially since they are derived from existing literature. The study's reliance on literature review and bioinformatics suggests a need for experimental validation to ascertain the functional relevance of these genes in bovine temperament. It would be beneficial to reference studies that have performed relevant experiments, detailing and discussing their outcomes to substantiate the role of these genes.

4.         Phenotypic Records and Genetic Analysis: The paper should provide an overview of the phenotypic markers commonly utilized in temperament research. Moreover, to capture a more extensive spectrum of genetic variations, consideration should be given to incorporating genome-wide association studies (GWAS) or whole-genome sequencing analyses.

5.         Interaction Network Analysis: Leveraging the STRING database for constructing the interaction network is a sophisticated approach that sheds light on the intricate genetic interplays linked to bovine temperament. Nonetheless, reliance on databases like STRING might introduce biases due to potentially outdated or incomplete annotations. It is recommended to undertake functional studies to demystify the roles of the Kelch family genes and SST, which appear pivotal in the temperament network.

6.         Biological Pathways: The paper adeptly discusses the involvement of the identified genes in biological pathways such as AMPA receptors and Hormones, enriching our understanding of the genetic underpinnings of temperament. However, the current analysis may not fully encapsulate the intricate and polygenic nature of temperament traits, suggesting a need for a more encompassing approach.

7.         Inclusion of Environmental Factors: While the review places significant emphasis on genetic factors, it somewhat overlooks the pivotal role of environmental influences on bovine temperament. Integrating environmental factors into the analysis is essential for a holistic understanding of the multifaceted determinants of temperament.

8.         Specificity of Gene Functions: For further enhancement, the paper should delve deeper into the specific functions of the identified genes within the context of bovine temperament, providing a clearer mechanistic understanding of their roles and interactions.

In summary, the paper presents a well-structured systematic review and network analysis to identify candidate genes related to bovine temperament. However, it would benefit from experimental validation and a more integrated approach that considers environmental factors and broader genetic analysis.

Reviewer 2 Report

Comments and Suggestions for Authors

Very interesting analysis, which allows for a broader view of the processes in the body that may be related to temperament. The results show important signaling pathways, but also connections between genes, which may be an important clue for further research on these traits in cattle.

The results are very readable, the analysis of candidate genes divided into appropriate clusters is valuable.

My only comment refers to the entry in lines 65 and 99 - it should be Web of Science and not Science of Web (the correct entry is in Figure 1).

The work is based on a review of current knowledge on the genetic determinants of cattle temperament and on this basis an attempt was made to determine the relationships between them, especially in relation to the latest indicated candidate genes and the structure of the signaling pathways in which they are involved.

An original and valuable element of the work is the indication, based on the analysis of the body's processes, which may play a significant role in the formation, but also in the externalization of traits related to temperament. This supplements the current knowledge with a broader context than just determining the variability within genes and their connections with traits. It also indicates significant relationships between them, interaction. Such a broader view shows the possibilities of further in-depth research in this direction.

The research methodology, analysis of various platforms with publications, specific criteria for selecting publications, the use of genetic databases and bioinformatics tools are correct and do not require corrections.

The presented conclusions result clearly from the presented results and are fully adequate to the assumed goal of the work.
The publications used present the latest knowledge, are correctly selected and used in the publication. Figures and tables are legible, well-developed, and the figures are of good quality.

Reviewer 3 Report

Comments and Suggestions for Authors

In this systematic review the authors investigated numerous candidate genes for bovine temperament and underwent an interactive network analysis based on these. A literature study revealed 36 articles, which highlighted 102 candidate genes. Starting from these, an interaction network containing 113 nodes and 346 interactions was then built based on the STRING database. Further analysis of this network revealed relevant interactions between the candidate genes and pathways implicated in previously studied functions and traits, and novel candidate genes were identified. The employed methodology is adequate for this type of study and the level of attention to details, as well as the number of candidate genes are suitable. The methodology and results are described in sufficient detail and they are relevant in the field, having good potential to support future studies. Furthermore, the quality of English writing is fine and no issues were found.

Author Response

REVIEWER 3. No comments to answer, thank you for your revision.

Reviewer 4 Report

Comments and Suggestions for Authors

Manuscript ID: genes-3100023

Title: A Comprehensive Systematic Review Coupled with an Interacting Network Analysis Identified Candidate Genes and Biological Pathways Related to Bovine Temperament.

General comments: The authors implement a systematic review following the PRISMA statement to analyze candidate genes related to bovine temperament. As a result, they found the complexity of interconnected biological processes that regulate behavior and stress response in mammals. Resutls from this manuscript could provide a clue for further investigating candidate genes in the bovine.

Major concerns:

Before we can identify candidate genes linked with bovine temperament, we have to measure temperament. Thus, how to measure temperment is vital for investigae genetic basis underlying boving temperament. I strongly suggest the authors add some information about how to measure temperament.

Introduction section: The authors should provide some essential information about PRISMA and provide why PRISMA was implented instead of traditonal literature reivew.

Specific comments:  

Conclusions section can be concised. The core inforamtion is whats the new finding when use the PRISMA to investigate bove temperament

Round 2

Reviewer 4 Report

Comments and Suggestions for Authors

The authors have addressed all of my concerns.